# The Local Tumor Microbiome Is Associated with Survival in Late-Stage Colorectal Cancer Patients

Justine W. Debelius,[a,b] Lars Engstrand,[a] Andreas Matussek,[c,d,e] Nele Brusselaers,[a,f,g] James T. Morton,[h] Margaretha Stenmarker,[i,j,k] Renate S. Olsen[a,l,m]

[a]Centre for Translational Microbiome Research, Department of Microbiology, Tumor, and Cell Biology, Karolinska Institutet, Solna, Sweden
[b]Department of Epidemiology, Johns Hopkins Bloomberg School of Public Health, Baltimore, Maryland, USA
[c]Laboratory Medicine, Jönköping Region County, Department of Clinical and Experimental Medicine, Linköping University, Jönköping, Sweden
[d]Division of Laboratory Medicine, Institute of Clinical Medicine, University of Oslo, Oslo, Norway
[e]Department of Microbiology, Division of Laboratory Medicine, Oslo University Hospital, Oslo, Norway
[f]Global Health Institute, Antwerp University, Antwerp, Belgium
[g]Department of Head and Skin, Ghent University, Ghent, Belgium
[h]Biostatistics and Bioinformatics Branch, Eunice Kennedy Shriver National Institute of Child Health and Human Development, National Institutes of Health, Bethesda, Maryland, USA
[i]Futurum/Department of Pediatrics, Jönköping Region County, Jönköping, Sweden
[j]Department of Clinical and Experimental Medicine, Linköping University, Linköping, Sweden
[k]Institute of Clinical Sciences, Department of Paediatrics, Sahlgrenska Academy at the University of Gothenburg, Gothenburg, Sweden
[l]Pathology Laboratory, Department of Laboratory Medicine, Jönköping Region County, Jönköping, Sweden
[m]Department of Pathology, Division of Laboratory Medicine, Oslo University Hospital, Norwegian Radium Hospital, Oslo, Norway

**ABSTRACT** The gut microbiome is associated with survival in colorectal cancer. Single organisms have been identified as markers of poor prognosis. However, *in situ* imaging of tumors demonstrate a polymicrobial tumor-associated community. To understand the role of these polymicrobial communities in survival, we conducted a nested case-control study in late-stage cancer patients undergoing resection for primary adenocarcinoma. The microbiome of paired tumor and adjacent normal tissue samples was profiled using 16S rRNA sequencing. We found a consistent difference in the microbiome between paired tumor and adjacent tissue, despite strong individual microbial identities. Furthermore, a larger difference between normal and tumor tissue was associated with prognosis: patients with shorter survival had a larger difference between normal and tumor tissue. Within the tumor tissue, we identified a 39-member community statistic associated with survival; for every $\log_2$-fold increase in this value, an individual's odds of survival increased by 20% (odds ratio survival 1.20; 95% confidence interval = 1.04 to 1.33). Our results suggest that a polymicrobial tumor-specific microbiome is associated with survival in late-stage colorectal cancer patients.

**IMPORTANCE** Microbiome studies in colorectal cancer (CRC) have primarily focused on the role of single organisms in cancer progression. Recent work has identified specific organisms throughout the intestinal tract, which may affect survival; however, the results are inconsistent. We found differences between the tumor microbiome and the microbiome of the rest of the intestine in patients, and the magnitude of this difference was associated with survival, or, the more like a healthy gut a tumor looked, the better a patient's prognosis. Our results suggest that future microbiome-based interventions to affect survival in CRC will need to target the tumor community.

**KEYWORDS** 16S rRNA sequencing, colorectal cancer, microbiome, cancer survival, tumor microbiome

Address correspondence to Renate S. Olsen, resols@ous-hf.no, or Justine W. Debelius, justine.debelius@jhu.edu.

The authors declare no conflict of interest.

Globally, colorectal cancer (CRC) is the second most common cause of cancer-related death and CRC-related mortality has been increasing since 2000 (1, 2). One potential area of research in CRC survival is the gut microbiome. In a healthy gut, the

intestinal microbiome contributes to homeostasis through epithelial cell renewal, maintaining gut barrier integrity, and immune modulation (3, 4). However, CRC patients have demonstrated a consistently altered gut microbiome compared to healthy controls, including a higher relative abundance of organisms more commonly found in the oral cavity (5, 6). Meta-analyses using targeted analyses show high levels of *Fusobacterium nucleatum* in tumor tissue are detrimental to survival (7, 8).

Fewer studies have explored the relationship between the gut microbiome and CRC prognosis using untargeted sequencing. Untargeted techniques can better characterize the bacterial community, and the ways in which potentially pathogenic organisms might interact with a host's unique, stable microbiome (9–11). *In situ* microscopy shows that tumor tissue is colonized by a polymicrobial biofilm, including *Fusobacteria*, *Proteobacteria*, *Bacteroidetes*, and *Lachnospriaceae*; monoculture biofilms have not been observed (12). Biofilms are also frequently localized to tumors, and paired normal tissue samples are rarely colonized, suggesting a localized effect and potential difference between tumor and adjacent tissue (12).

Previous untargeted studies of the gut microbiome and colorectal cancer survival have either focused exclusively on the tumor tissue (13) or have treated the tumor and adjacent normal tissue as identical (14). Paired biopsy studies provide clues about whether local or global regulation of the microbiome drives tumorigenesis, although many paired studies have failed to account for survival (12, 15–19) and, in some cases, struggled to characterize the microbiome due to technical (19) or analytical (13–17) issues.

To address the gaps in knowledge, we monitored 101 late-stage CRC patients recruited from a hospital in southern Sweden who underwent surgical resections of primary adenocarcinoma between 1997 and 2017. Patients were categorized into short- or long-term survivors based on their relapse-free survival (<2 years or ≥5 years, respectively). We examined the relationship between the microbiome of colorectal tumors and adjacent normal tissue and survival, accounting for clinical covariates.

## RESULTS

In our nested case-control study of late-stage colorectal cancer patients, the 51 long-term survivors were more likely to be younger, male, and healthier compared to the 50 short-term survivors (see Table S1 in the supplemental material). The short-term survivors presented with metastatic tumors and lower differentiation than long-term survivors, and fewer received radical surgery. We found that age, tumor-node-metastasis (TNM) stage, and tumor differentiation were strong predictors of long-term survival (Table 1). Individuals aged between 70 and 74 years were 14 times more likely to be short-term survivors (odds ratio [OR] = 14.24; 95% confidence interval [CI] = 1.21 to 167.40) than those younger than 60. TNM stage IV was associated with an almost 50 times higher risk of being a short-term survivor (OR = 49.32; 95% CI = 5.86 to 415.12) compared to TNM stage III (Table 1).

After sequencing, quality filtering, and denoising to amplicon sequence variants (ASVs), we retained 202 paired tumor and adjacent normal tissue samples. The broad patterns in the overserved microbiome reflect those seen in a previous study of Swedish adults (see Fig. S1) (20). We found the patient was the strongest predictor of microbiome composition and that an individual's paired samples were more similar to each other than the same type of tissue from patients matched for cancer stage and anatomical location (see Fig. S2), reflecting what appears to be a common pattern in CRC patients and beyond (10, 18, 21).

**The microbiomes of tumor and normal tissue differ.** To address individual microbial identities, we applied a subject-aware compositional tensor factorization (CTF) ordination technique (22). This analysis projects high dimensional microbiome data into a three-dimensional ordination space, relating samples and their component features (22). We did not find a statistically significant association between a subject's position in CTF space and survival (unadjusted permanova $R^2$ = 0.012; $P$ = 0.296, 999

**TABLE 1** Risk factors for short-term survival

| | OR (95% CI)[a] | | |
| | Crude risk | Adjusted risk | |
| Characteristics | | Model 1 | Model 2 |
|---|---|---|---|
| **Patient characteristics at surgery** | | | |
| Age, yrs | | | |
| <60 | 1.00 (ref) | 1.00 (ref) | 1.00 (ref) |
| 60–69 | 0.87 (0.24–3.09) | 2.45 (0.26–22.72) | 2.59 (0.28–24.38) |
| 70−74 | 2.40 (0.65–8.81) | 12.55 (1.06–149.26) | 14.24 (1.21–167.40) |
| ≥75 | 1.96 (0.56–6.91) | 8.68 (0.79–95.19) | 10.55 (0.99–112.75) |
| Sex | | | |
| Female | 1.00 (ref) | 1.00 (ref) | 1.00 (ref) |
| Male | 0.76 (0.35–1.67) | 0.47 (0.14–1.56) | 0.44 (0.13–1.41) |
| ASA score | | | |
| I (healthy) | 1.00 (ref) | 1.00 (ref) | 1.00 (ref) |
| II (mild) | 0.80 (0.32–2.02) | 2.29 (0.45–11.78) | 2.69 (0.56–12.96) |
| III-IV (severe or worse) | 2.01 (0.65–6.19) | 4.10 (0.60–27.92) | 4.99 (0.79–31.45) |
| Preoperative treatment | | | |
| None | 1.00 (ref) | 1.00 (ref) | |
| Radiotherapy | 1.17 (0.74–1.84) | 0.71 (0.12–4.15) | |
| **Tumor characteristics** | | | |
| Localization | | | |
| Colon right | 1.00 (ref) | 1.00 (ref) | 1.00 (ref) |
| Colon left | 0.47 (0.16–1.32) | 0.78 (0.16–3.82) | 0.76 (0.16–3.63) |
| Rectum | 0.72 (0.28–1.84) | 2.03 (0.33–12.63) | 1.61 (0.36–7.21) |
| Mucinous cancer | | | |
| No | 1.00 (ref) | 1.00 (ref) | |
| Yes | 0.83 (0.24–2.93) | 0.50 (0.05–5.39) | |
| TNM stage | | | |
| III | 1.00 (ref) | 1.00 (ref) | 1.00 (ref) |
| IV | 10.8 (3.68–31.72) | 44.67 (5.53–360.63) | 49.32 (5.86–415.12) |
| Grade of differentiation | | | |
| Low | 1.00 (ref) | 1.00 (ref) | 1.00 (ref) |
| Medium | 0.20 (0.07–0.54) | 0.23 (0.05–0.98) | 0.24 (0.06–1.00) |
| High | 0.21 (0.05–0.97) | 0.09 (0.01–1.24) | 0.10 (0.01–1.27) |
| **Surgical characteristics** | | | |
| Period of surgery | | | |
| 1997−2005 | 1.00 (ref) | 1.00 (ref) | 1.00 (ref) |
| 2006−2010 | 0.54 (0.21–1.37) | 0.44 (0.10–1.92) | 0.44 (0.10–1.89) |
| 2011−2017 | 0.59 (0.21–1.65) | 1.19 (0.22–6.47) | 1.08 (0.22–5.36) |
| Radical operation | | | |
| No | 1.00 (ref) | 1.00 (ref) | 1.00 (ref) |
| Yes | 0.05 (0.01–0.41) | 0.13 (0.01–1.51) | 0.12 (0.01–1.34) |

[a]Model 1 values are adjusted for all variables. Model 2 values are adjusted for all variables except for preoperative treatment and mucinous cancer. ref, reference.

permutation; see also Fig. S3 and Table S2). However, we found differences between normal and tumor tissue in CTF space. Paired samples from the same individual showed consistent, directional differences, primarily along principal component (PC) 2 and PC 3 (Fig. 1A to D; permutative paired sample $t$ test, $P = 0.001$, with 999 permutations for all PCs).

Given evidence of consistent, community-level changes in the microbiome between the tissue types, we looked for features, which might be driving these differences. We used an individual-aware differential ranking technique (DR), which first ranked the features with the greatest differences associated with tissue type, and then we selected a subset of these features to build an additive log ratio (ALR), a summary of taxa which likely describe the difference (Fig. 1E; see also Table S3) (23–25). We found that tumor tissue was associated with a higher relative abundance of *Fusobacteria*, *Porphyromonas*, *Granulicatella*, and *Campylobacter* at the expense of members of genus *Blautia* and *Ruminococcus*. Tumor

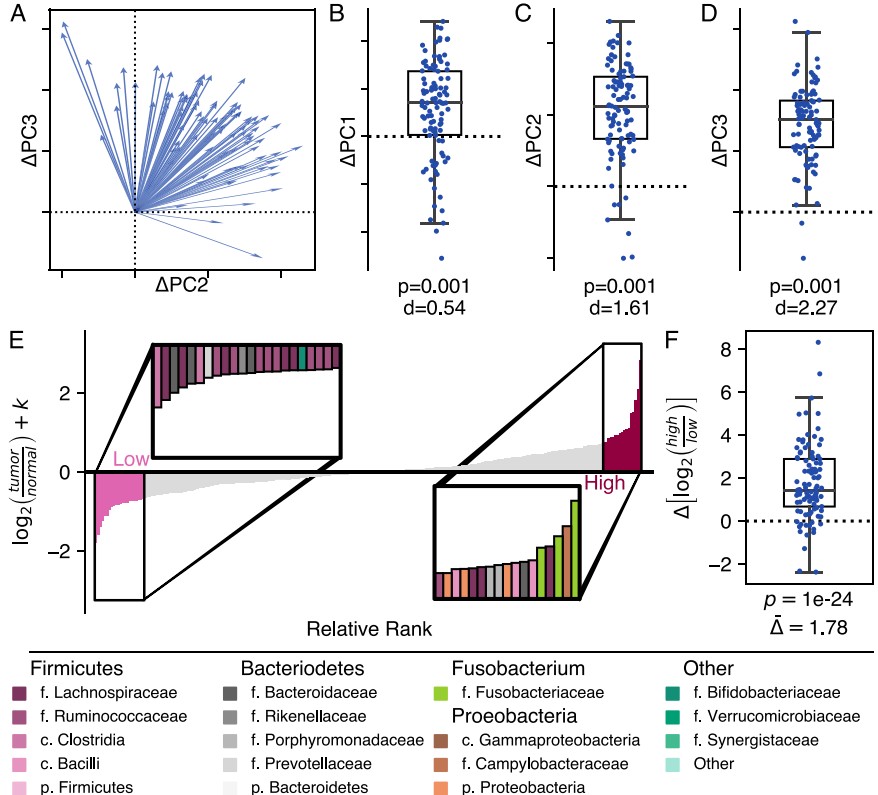

**FIG 1** There is a difference in the microbiome between tumor and normal tissues. We found a global pattern separating tumor and normal tissue, which can be seen in CTF ordination space. (A) Plotting the change between normal and tumor tissue in PC 2 and PC 3 as a vector with normal tissue as the center demonstrates a clear directional pattern. The difference between normal and tumor tissue can also be observed along individual components: PC 1 (B), PC 2 (C), and PC 3 (D). Ticks and dashed zero-lines along PC 2 (C) and PC 3 (D) match the two-dimensional axes in panel A. All boxplots are shown with a Cohen's d effect size statistic for a one-sided $t$ test and $P$ values from a permutative one sample $t$ test, with 999 permutations. (E) Differential ranking of 300 abundant features identified normal tissue-associated features (light pink) and tumor tissue-associated features (dark pink). The inset shows selected features in each group, colored by family; colors are defined in the legend. (F) Change in the ALR between normal and tumor tissue. The coefficient from a linear mixed-effect model comparing the change based on tissue type was determined.

tissue had a 1.78 (95% CI = 1.50 to 2.18, $P < 1 \times 10^{-12}$) log$_2$-fold increase in the features selected by DR compared to normal tissue, suggesting a tissue-specific signature (Fig. 1F).

Since these observations conflict with the existing literature, we reanalyzed previously published data to confirm our findings (21). We first determined the paired samples from a single individual were more similar to each other than any other samples in the replication cohort (see Fig. S4A and B; $P = 0.001$, 999 permutations). We then applied the global test used in the previously published paper to both our cohort and the replication cohort, which interrogated whether there was a statistically significant, global separation between the two tissue types (see Table S3 and Fig. S4C). In line with previous work (21), we did not find a statistically significant, global separation, measured by a permanova in either data set. However, when we applied the sample subject-aware CTF technique on the replication set, we found a clear, statistically significant difference along all three PCs (permutation $P = 0.001$, 999 permutations; see also Fig. S4D to G). We then tried replicating the tissue associated ALR in the validation cohort (see Fig. S4F). Tumor tissue in the validation cohort had a 1.70 (95% CI = 0.66, 3.00; $P = 0.003$) log$_2$-fold increase in the features selected by DR compared to normal tissue (see Fig. S4H).

Our results therefore suggest that while an individual's microbial identity plays a strong role in shaping the microbiome, subject-aware comparisons are associated with a consistent, reproducible difference in the microbiome on and off tumors in colorectal cancer.

**Differences between normal and tumor associated microbiome are associated with survival.** Since we saw consistent differences between tumor and normal tissue, we wondered whether there might be a relationship between the magnitude of the difference between tissue types and survival. Using traditional dissimilarity-based beta diversity, we found that tumor and normal tissue were more similar in long-term survivors than short-term, a difference primarily driven by greater change in abundant features (see Table S5). In addition, long-term survivors showed a greater change along PC 2 in our CTF ordination compared to short-term survivors (Cohen's d = 0.40, $P$ = 0.016, 999 permutations; Fig. 2). This suggested enough of a community-level change in the microbiome to motivate looking for features which might explain the differences.

Therefore, we applied a subject-aware differential ranking technique looking at the interaction between tissue type and survival to further refine the features (Fig. 2E to G). We used an interaction model to identify features that changed in tumor tissue based on survival group. Based on the tissue associated taxa associated with long-term survival, we defined an ALR where tumor tissue was associated with a higher relative abundance of ASVs from genus *Fusobacterium*, *Campylobacter*, and *Escherichia/Shigella* compared to ASVs from families *Lachnospiraceae* and *Rumminococeae* (see Table S6) (25). We found members of genus *Butyricicoccus*, *Roseburia*, and *Streptococcus* associated with both normal and tumor tissue. There was a higher relative ratio of the tumor associated organisms in tumor tissues from both short- and long-term survivors, and the overall ratio was significantly higher in short-term survivors (Fig. 2F). However, the magnitude of the difference between normal and tumor tissue did not differ between the short- and long-term survivors.

In contrast, the interaction term identified a set of taxa, which were significantly different between the tissue types in short-term survivors but not among long-term survivors (Fig. 2G; see also Table S7) (25). Once again, we found tumor tissue in short-term survivors to be strongly associated with an ASV from *Fusobacteria* and as well as a few members of family *Veillonellaceae*, although again, there were not clear taxonomic patterns observed in the rest of the ASVs used to construct our taxonomic ratio. These results indicate the survival-associated changes in the microbiome may be largest in tumor tissue and help to identify a specific set of organisms responsible for these changes.

**The tumor microbiome is associated with survival.** Based on our observation that differences in tissue types were more pronounced in short-term survivors, and since past work focused on tumor tissues, we chose to further interrogate the tumor-specific microbiome. We applied robust principal component analysis (rPCA), an ordination technique designed for microbiome data which combines sample and ASV information into the same plot (22). Our ordination showed separation in the microbiomes between short- and long-term survivors (Fig. 3A and D). After adjustment for confounders and both PCs, patients with larger values for PC 1 had 3.5 lower odds (OR = 0.29; 95% CI = 0.08 to 0.97) of short-term survival, while those with higher values for PC 2 were five times less likely to be short-term survivors (OR = 0.19; 95% CI = 0.05 to 0.80). Individuals in the quadrant defined by these two extremes in the data were at least 7.5 times more likely to survive than any other group in the ordination (see Fig. S5).

We found 37 features associated with separation along PC 1. To the left of PC 1, we found members of the genera *Fusobacterium*, *Parvimonas*, and *Porphyromonas*, as well as other common oral genera such as *Gemella* and *Dialster* (Fig. 3B). In contrast, higher values along PC 1 (to the right) were correlated with more common gut taxa, including members of families *Lachnospiraceae* and *Rumminococceae*. We defined the $\log_2$-fold ratio between the organisms separating PC 1 as a tumor survival index (see Table S8) (25). For every 2-fold increase in this index in tumor tissue, the odds of survival increased by 20% (adjusted OR = 0.80; 95% CI = 0.67 to 0.96). There were no clear patterns in the taxa separating along PC 2, beyond the association between *Escherichia/Shigella* and short-term survival, although there was a significant relationship between these selected taxa and survival (OR = 0.64; 95% CI = 0.41 to 0.98 for every $\log_2$ increase).

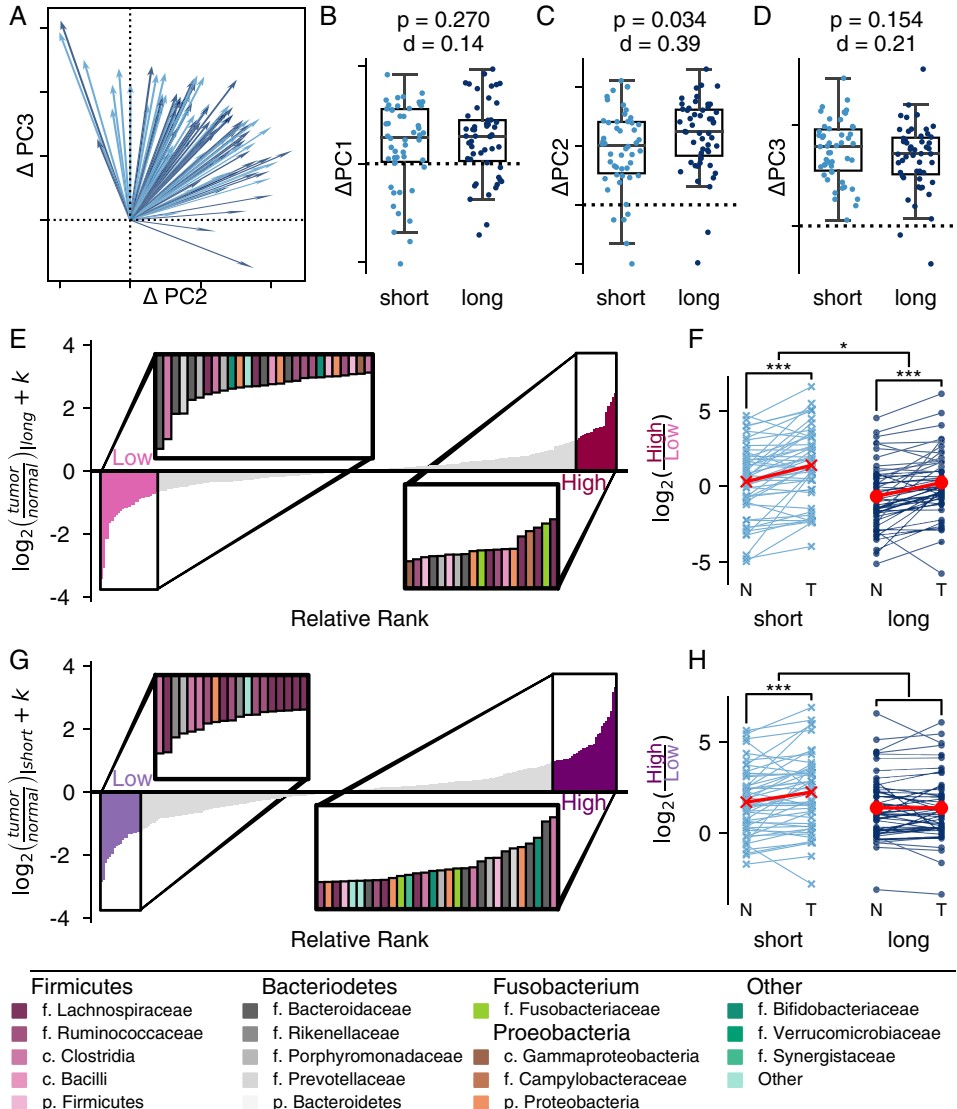

**FIG 2** The magnitude of the difference between tumor and normal tissue is associated with survival. (A) In two dimensions, the change along PC 2 and PC 3 is visualized as a vector from normal to tumor tissue. Short-term survivors (<2 year) are indicated in light blue. Long-term survivors (≥5 years) are indicated in dark blue. (B to D) The corresponding relationships can be visualized along the individual components: PC 1 (B), PC 2 (C), and PC 3 (D). Ticks along PC 2 (C) and PC 3 (D) match the two-dimensional axes in panel A. All boxplots are shown with a Cohen's d effect size and a *P* value from a permutative Welch's *t* test with 999 permutations, comparing the two survival groups. (E to H) A differential ranking model was fit to consider the interaction between survival and tissue. The ranks associated with tumor tissue in long-term survivors (E and F) and tumor tissue in short-term survivors (G and H) (interaction) are shown. (E and F) Relative ranks associated with the model. Insets highlight the ASVs associated with the extremes of each group. Taxonomic assignments are provided in the legend. (F and H) Additive $log_2$ ratio associated with the ranks. Paired differences are connected by a line between normal (N) and tumor (T) tissues. The effect was modeled using a linear mixed-effects model, treating the individual as random (*, $P < 0.05$; **, $P \leq 0.01$; ***, $P \leq 0.001$).

## DISCUSSION

Our results show a clear and consistent difference between normal and tumor tissue once we had accounted for individual microbiome effects. Across all patients, tumors carried a higher proportion of ASVs mapped to genus *Fusobacterium*, *Gemella*, *Dialster*, and *Campylobacter* at the expense of genera such as *Blautia* and *Allistipes*. The tumor associated microbiome features reflect organisms found more commonly in CRC patients compared to healthy controls, whereas the organisms associated with normal tissue belong to clades commonly associated with short-chain fatty acids and

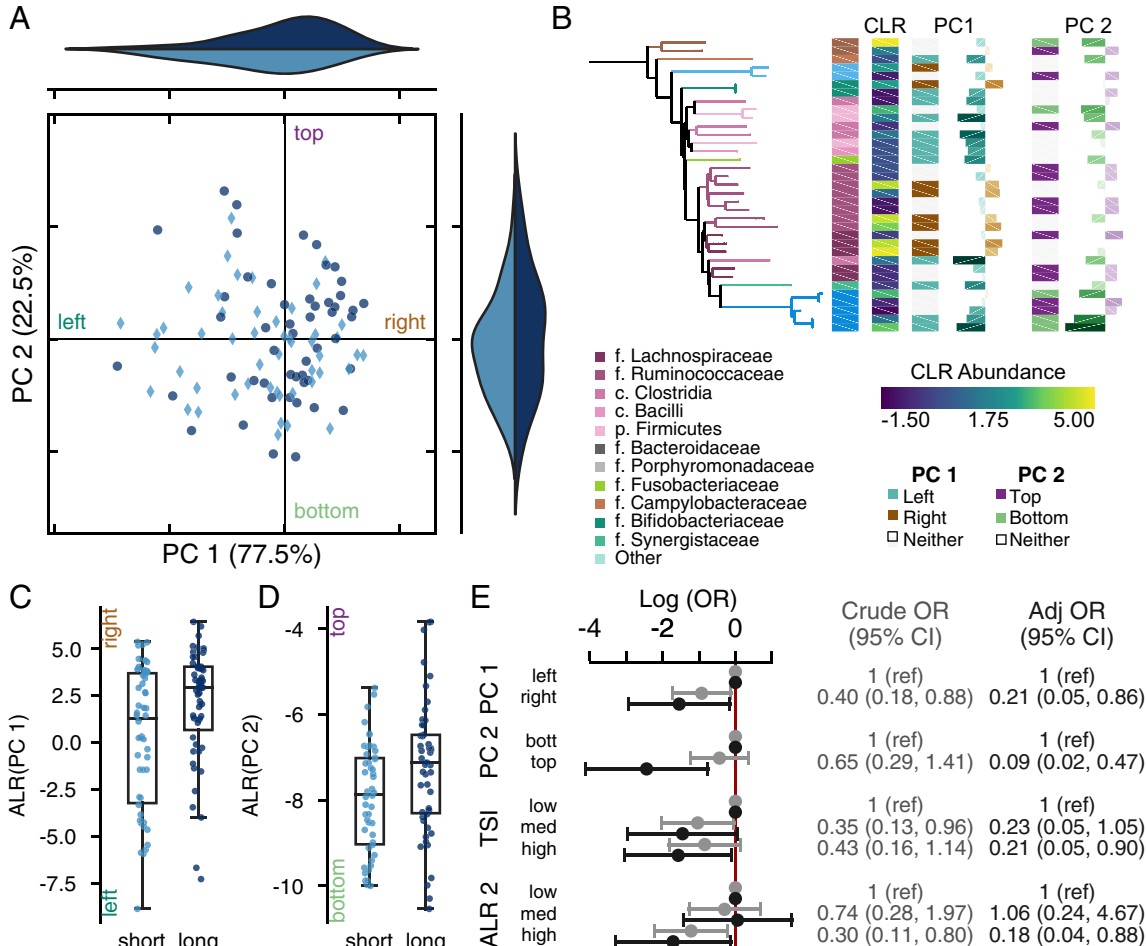

**FIG 3** The tumor-associated microbiome is associated with survival. (A) Robust principal component analysis (rPCA) ordination as indicated by short-term (light) or long-term (dark) survival. Marginal axes show the distribution of points along each PC. The ordination is centered at the median distribution of points in each axis. (B) Phylogenetic tree showing the ASVs with PC 1. The tips and first heatmap colors indicate taxonomic information. Heatmaps from left to right show the taxonomic assignment, the mean central log ratio (CLR) relative abundance (viridis), the mean difference in CLR between long- and short-term survivors; whether the feature was used in the additive log ratio (ALR) calculation for PC 1 (teal, left; brown, right); the feature loading along PC 1; whether the feature was used in the PC 2 ALR calculation (green, bottom; purple, top); and the feature loadings along PC 2, coordinates to the top on the left. (C and D) Boxplot of the ALR of most extreme ranked taxa along selected based on PC 1 (C) and PC 2 (D). Axes are labeled to indicate the directionality of the log ratio calculation; the label colors matchthose in panel B. (E) Log odds of short survival based on separation along the median of the PCs or grouping in the ALR shown in light gray values are crude; those in dark gray are adjusted for age, sex, surgery year, tumor location, ASA score, differentiation grade, and TNM stage. PC weights are adjusted for the position along the other PC; ALRs are not coadjusted.

widely believed to be beneficial (5, 6, 26–28). Further, we are among the first to show that the magnitude of the difference between the normal and matched cancer tissue can be associated with prognosis. Our differential ranking analysis identified a set of 38 ASVs, which changed between the tumor and normal tissue in short-term survivors, but not long-term survivors. This suggests that survival may be associated with localized changes in the microbiome.

We are among the first to report differences between tumor and normal tissues in paired samples, let alone an association between the degree of dissimilarity and survival. Drewes et al. (12) demonstrated clear difference between paired tumor and normal tissue samples using microscopy, although their 16S analysis did not explicitly test paired samples. These results seemingly conflict with much of the existing literature (15–18, 21). Several previous studies reported no difference in the microbiome between the two tissue types, let alone an intraindividual difference associated with survival. As in past studies, we observed and described a strong intraindividual similarity:

a personal microbial signature is a normal feature of the microbiome seen in a variety of settings, including population-based studies (10), dietary interventions (29), and among CRC patients (15–18, 21). However, unlike prior work, the statistical models we selected accounted for this strong intraindividual similarity. Our work suggests that model selection is critical: the difference is not observed with methods that do not account for the subject-specific variation and instead look for global changes. We demonstrate that reanalysis of prior publications using subject-aware methods replicates the patterns we found: a difference between the tissue types, despite strong individual microbiome signatures. Our results indicate that the tumor-specific microenvironment, rather than the overall microbiome, is important for understanding CRC pathology. At a minimum, future sequencing survey studies will need to account for tissue-specific effects in their analysis, and studies treating tumor and nontumor biopsy samples as identical may need to check for biases.

Based on the difference in the microbiome between tissue types, we specifically focused on the relationship between the tumor microbiome and survival. Two previous studies have explored the relationship between the tumor microbiome and survival using untargeted sequencing. In that study of 67 Irish patients, Flemer et al. defined microbiome groups using a noncompositional abundance-based clustering approach (14). These researchers found a higher relative abundance of a cluster defined by members of the genera *Bacteriodetes*, *Blautia*, *Roseburia*, and *Rumminococus*, as well as an unclassified member of the family *Lachnospiracae*, was associated with shorter survival, while a higher abundance of a cluster characterized by *Streptococcus*, *Fusobacterium*, and unclassified family *Enterobacteraceae* was associated with longer survival. These groupings are contradictory to the features associated with survival in our tumor tissue results. In contrast, our tumor survival index, defined by an ALR of features along PC 1, showed a decrease in the relative abundance in *Fusobacterium* compared to the relative abundance of genera like *Blautia* and *Roseburia*. It is likely that this disagreement is due, at least in part, to differences in methods used for differential abundance (23, 30). Our results are more in line with results from a Chinese cohort (13). In that study, a higher untransformed relative abundance of genus *Fusobacterium* or a higher relative abundance of reads mapped to "*Bacteriodetes fragilis*" was associated with an increased hazard of death, while a higher relative abundance of genus *Faecalibacterium* was protective. We find similar trends in our tumor survival index, where short-term survival was associated with ASVs mapped to genus *Fusobacterium* and a *Bacteriodetes* ASV, while longer survival was associated with *Faecalibacterium*. Our results and those of the Chinese cohort suggest that a more normal (gut-like) microbiome is associated with long-term survival, while a more disrupted (oral) microbiome led to a poor prognosis.

Our conclusions are supported by our nested case-control design, which helps establishing temporality: changes in the local tumor microbiome at the time of surgery are associated with future outcomes, increasing the probability that the observation is a real phenomenon, rather than a change in the microbiome in response to disease state. Our analysis used statistically appropriate methods, which accounted for analytical challenges in describing the microbiome, decreasing the possibility of false positives, especially among the identified taxa (23, 31). Our analysis has also addressed confounders, which may affect the microbiome and survival, including the strong individual microbiome signature.

However, our study has some limitations. First, our results focus on late-stage cancer patients in northern Europe and therefore may not be broadly generalizable. There are reports of differences in the tumor microbiome between early- and late-stage CRC patients (32) and differences in healthy microbiomes between countries (33). However, past work has suggested that CRC is characterized by a set of organisms similar to the ones we identified, and our work overlaps with the results of a Chinese cohort, despite methodological differences (5, 6, 13). We were unable to find a suitable publicly available cohort with sufficient metadata to validate our tumor survival index; the features we identified may be specific to our cohort rather than able to predict survival in a

broader population of late-stage CRC patients. Finally, we profiled the microbiome using 16S rRNA sequencing, with all the assumptions, benefits, and limitations of the technique. Our work is predicated on the assumption that phylogenetic similarity correlates to genetic and niche similarity. Without robust functional prediction and the ability to assemble genome units, we are limited in our mechanistic insight. However, our 16S sequencing is, in many cases, able to capture species- or subspecies-level resolution as the amplicon sequence variant ID, even if the name cannot be inferred accurately (34, 35).

In conclusion, we performed a nested case-control of the role of the microbiome in relapse-free survival following primary resection in late-stage CRC patients. We identified clear differences in the microbiome between normal and tumor tissue and that a larger difference between tissue types was associated with poor prognosis. We found the tumor microbiome was associated with survival. This suggests a need to focus microbiome-based interventions at the tumor-specific community rather trying to modify prognosis by changing the gut microbiome overall.

## MATERIALS AND METHODS

**Study population.** Patients were recruited from all consecutive CRC patients ($n$ = 540) who underwent surgical resection for primary colorectal adenocarcinoma at the Department of Surgery, Ryhov County Hospital, Jönköping Region County, Jönköping, Sweden, between 1997 and 2017. Patients with tumor-node-metastasis (TNM) stage III and IV cancer at the time of surgery who had matched biopsy specimens from normal and tumor tissue ($n$ = 116) were selected. Patient details, including demographic, surgical, pathological information, and survival outcomes were determined from a review of medical records.

The final study cohort included patients with paired, high quality microbiome samples ($n$ = 101). Fifteen individuals were excluded due to insufficient sequencing depth in the tumor ($n$ = 8) or normal ($n$ = 7) tissue sample. There was no difference in the survival status in the samples with insufficient sequencing depth. Included patients had matched tumor and normal tissue samples ($\geq$10 cm apart from tumor tissue). Our analysis included samples from 51 long-term ($\geq$5-year survival) and 50 short-term ($\leq$2-year survival) survivors.

The study was approved by the Regional Ethical Review Board in Linköping, Linköping, Sweden (98113, 2013/271-31). A written informed consent was obtained from each patient.

**Statistical analysis of patient characteristics.** A multivariable logistic regression was used to assess the predictive impact of the following patient-, cancer-, and treatment-related characteristics: age (categorized as <60, 60 to 69, 70 to 74, and $\geq$75 years), sex (female or male), American Society of Anesthesiologists physical status (ASA) score (I, healthy; II, mild; III and IV, severe [patients with V and VI scores were not eligible for surgery]), localization of the tumor (right colon, left colon, rectum), TNM stage (III or IV), grade of differentiation (from low differentiation to high differentiation, with the latter more closely resembling noncancer histology), radical surgery (yes or no); and period of surgery (1997 to 2005, 2006 to 2010, and 2011 to 2017). All results are expressed as odds ratios (ORs) and 95% confidence intervals (CIs), and the calculations were conducted with Stata MP14 (Stata Corp., College Station, TX).

**Microbiome sequencing.** Paired tumor and normal tissue samples were collected were collected during colorectal resection surgery. Tissue samples were frozen directly and stored at −80°C until use. Samples were processed as previously described (36). Briefly, DNA was extracted from tissue samples using physical and chemical lysis for extraction. The 16S rRNA amplicon library was amplified with 341F/805R primers (CCTACGGGNGGCWGCAG/GGACTACHVGGGTATCTAAT) using a program with 20 cycles (37). The samples were sequenced with a 2 × 300 approach using an Illumina MiSeq (San Diego, CA).

The demultiplexed reads were denoised using the DADA2 algorithm (v1.13.1) in R (38). After reads were demultiplexed and primers were trimmed, forward reads were trimmed to 265 nucleotides (nt) and reverse reads were trimmed to 225 nt; the error rate model was trained on 15% of the reads. Reads were joined with an at least 30-nt overlap, and anything shorter than 380 nt after joining was discarded. Taxonomic assignment was performed using the naive Bayesian classifier implemented in DADA2 against the Silva 128 database (39, 40). The ASV table from DADA2, taxonomy, and representative sequences were imported into QIIME 2 (v2020.11) for further processing (41). A phylogenetic tree was built using fragment insertion using the SEPP algorithm into the Silva 128 backbone with q2-fragment insertion (40–42). The table and sequences were filtered to exclude any ASV without phylum-level annotation or which could not be inserted into the phylogenetic tree.

**Microbiome community characterization. (i) Between-sample (beta) diversity.** For paired-sample analysis, we calculated unweighted UniFrac (43), weighted UniFrac (44), and binary Jaccard (45) distances and Bray-Curtis dissimilarity (46) on a feature table rarified to 2,500 sequences/sample (47). Aitchison distance was calculated on unrarefied data with a pseudocount of 1 (31, 48). Beta-diversity metrics were calculated using the q2-diversity plugin in QIIME 2 (41).

**(ii) Compositional tensor factorization ordination.** To account for subject-specific effects on ordinations, we used compositional tensor fraction (CTF) for paired samples using the Gemelli qiime2 plugin (0.7.0) (22). Features were filtered to exclude those present in fewer than 20 samples or with <100 total counts. The distance in CTF subject space was calculated as the Euclidean distance between subject

coordinates. The difference in intraindividual CTF space between normal and tumor tissue (ΔPC) were compared using the subject-state coordinates.

**(iii) Robust principal component analysis.** For each tissue type, we examined beta-diversity using a robust principal component analysis (rPCA) using the DEICODE algorithm (v0.2.4) (49). For a given sample set, we filtered filtering features present in <10% of tumor samples ($n = 10$) or with fewer than 10 total counts. The auto-rPCA function was used to select the appropriate number of principle components (PCs) for the data. The PCs were divided into quartiles and dichotomized along the median value.

Within tumor tissue, which showed a significant association between microbiome and our outcome, we selected features that might be associated with survival. Communality was calculated as the square root of the sum of squares across all PCs. Features with a communality value of at least 0.01 were selected as candidates for the additive log ratio (ALR) calculation ($n = 130$). A pseudocount of 1 was added before the ALR calculation. The ALR was calculated as the $\log_2$ ratio of features more extreme than the fourth quartile of samples over features more extreme than the first quartile. Continuous ALR values or ALR divided into tertials were used for regression.

**(iv) Differential ranking.** We performed hypothesis generating differential abundance testing between tumor and normal tissue using a modified differential ranking (DR) technique (23, 24). We first filtered the table to remove any feature with a relative abundance of <1/1,000 in fewer than 10% of samples, leaving 243 features for testing. We then used a modified Bayesian method for DR testing. ASV counts were modeled through a negative binomial process. We started with naive priors of between a 0-fold and a 5-fold change in a ASV and fit the model using 4,000 iterations. The data were fit to a linear mixed effects model using subject as a random intercept, modeling either for tissue or for the intersection between tissue and survival. Modeling was done with pystan (v3.4.0) within the QIIME 2 2021.11 conda environment (50, 51).

We used the ranks to identify "extreme" features. Starting from the feature with the strongest signal associated with each possible value for a variable (e.g., normal versus tumor tissue and short- versus long-term survival), we added features until every tissue sample contained at least one of the extreme features. A pooled ALR was calculated as the sum of all normal tissue associated features over the tumor-associated features.

**Replication cohort.** We performed replication analysis on previously published data (21). Paired-end reads were downloaded from the European Nucleotide Archive (accession PRJEB47197); metadata was extracted from Table S1 from a study by Cronin et al. (52) (sheet name "Flemer et al., 2017 metadata"). Paired-end reads were imported into QIIME 2 for processing using a manifest format (41). The data were denoised using the q2-dada DADA2 implementation with default parameters aside from trim lengths (38). We trimmed the first 15 nt off the forward and reverse reads and then trimmed the forward reads to 240 nt and the reverse reads to 225 nt before denoising.

We identified 25 participants with paired tumor and normal tissue samples, six of whom had more than two samples. In those cases, we randomly selected the second sample for analysis, since no additional information was available. We calculated Bray-Curtis dissimilarity and Jaccard distance on a feature table rarefied to 1,000 sequences per sample using the qiime2 diversity plugin (41, 45, 46). We also performed CTF ordination on the replication data (22). The table was filtered to exclude features present in fewer than five samples or with fewer than 100 total counts. The changes along PCs 1, 2, and 3 were calculated, as described for the main cohort.

We also worked to validate the additive log ratio between tissue types. We clustered the representative sequences from the validation cohort against the representative ALR sequences (see Table S4) at 98% identity using the closed reference approach implemented in vsearch (q2-vsearch; vserach v2.7.0) (25, 53). We added a pseudocount of 1 and calculated the additive log ratio based on the groups from Table S4.

**Statistical analysis.** Paired distances were extracted as the distance between an individual's tumor and adjacent normal tissue. Interindividual distance was compared to the interindividual distance to samples of the same tissue type, anatomical location, and survival group with a permutative two sample $t$ test with 999 permutations.

The global difference in centroid between normal and tumor tissue, we applied a permutational multivariate analysis of variance (PERMANOVA) with 999 permutations in scikit-bio (v0.5.6) (54). Associations with per-subject CTF coordinates were checked by calculating the Euclidean distance between tissue samples and applying a PERMANOVA test with 999 permutations in scikit-bio (v0.5.6) (54). The change between tissue types in CTF coordinate space were modeled with a paired sample $t$ test was used to determine whether there was a global difference between tumor and normal tissue along either PC; the effect of change on survival was compared using a permutative Welch's $t$ test looking at the difference between groups with 999 permutations. ALR interactions were evaluated using a linear mixed effects model with individual as the grouping factor.

Survival was modeled using logistic regression. Models were fit using a crude (unadjusted) model and a model adjusted for age, sex, ASA score, tumor location, surgery period, TNM stage, radical surgery, and differentiation grade. Modeling was performed using statsmodels (v0.11.1), scipy (v1.4.1), and numpy (v1.18.5) in python (v3.6) (55–57).

For all analyses performed, a $P$ value of 0.05 was considered statistically significant.

Figures were plotted using with matplotlib (v3.2.2) and seaborn (v0.10.1) The dendrogram was plotted using Empress (q2-empress v0.0.1-dev, commit b705358) (58); three dimensional ordinations were rendered using Emperor (v1.0.3) (59). Taxonomic colors come from the microshades colorblind friendly palette (60). Figures were assembled in Illustrator 2021 (Adobe, Inc., San Jose, CA).

**Data availability.** Raw sequencing data and corresponding metadata are deposited in ENA under accession number PRJEB57580. Precalculated feature tables and metadata are also available through GitHub on at https://github.com/ctmrbio/crc-survival (v2.0 https://doi.org/10.5281/zenodo.7690117). Representative sequences and index tables for each of the ALR sets are deposited on Zenodo (https://zenodo.org/record/7696883) (25).

Tables were generated with code from https://github.com/ctmrbio/Amplicon_workflows.

Analysis notebooks for these data can be found on Github at https://github.com/ctmrbio/crc-survival; the revised manuscript is based on version 2.0 (https://doi.org/10.5281/zenodo.7690117) (61).

## SUPPLEMENTAL MATERIAL

Supplemental material is available online only.

**SUPPLEMENTAL FILE 1**, PDF file, 1.5 MB.

## ACKNOWLEDGMENTS

This study was funded by Futurum-Academy for Healthcare, Region Jönköping County, Sweden (grants FUTURUM-933436 and FUTURUM-809281), as well as a center grant from Ferring Pharmaceuticals for the establishment of the Centre for Translational Microbiome Research. J.T.M. was funded by the intramural research program of the Eunice Kennedy Shriver National Institute of Child Health and Human Development.

The funders were not involved in the development, analysis, or interpretation of the study.

We thank the Department of Surgery, County Hospital Ryhov, for the collection of tissue biopsy specimens. We thank the lab core at Centre for Translational Microbiome Research for support in extracting, processing, and sequencing the tissue samples. We are also grateful to Cameron Martino, Marcus Fedarko, and Kalen Cantrell for their rapid responses to bug reports and feature requests for the gemelli and empress qiime2 plugins. We appreciate the insightful conversations with Lorenzo Servitje on the nature of ordination space and alternative ways to discuss dimensionality reduction.

R.S.O., M.S., L.E., and A.M. designed the study. R.S.O. collected the tissue samples, performed DNA extraction, and reviewed medical records. J.W.D. prepared the data. J.W.D. performed the bioinformatic analysis. J.W.D. and N.B. analyzed the data with advice from J.T.M., J.W.D. drafted the manuscript. All authors reviewed and approved the final manuscript.

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
