## [Reviewer comments · Microbiology Spectrum]

Microbiology Spectrum

The local tumor microbiome is associated with survival in late-stage colorectal cancer patients

Justine Debelius, Lars Engstrand, Andreas Matussek, Nele Brusselaers, James Morton, Margaretha Stenmarker, and Renate Olsen

Corresponding Author(s): Justine Debelius, Johns Hopkins University Bloomberg School of Public Health

Review Timeline:

Submission Date:	December 9, 2022
Editorial Decision:	January 3, 2023
Revision Received:	March 6, 2023
Accepted:	March 27, 2023

Editor: Zhenjiang Xu

Reviewer(s): Disclosure of reviewer identity is with reference to reviewer comments included in decision letter(s). The following individuals involved in review of your submission have agreed to reveal their identity: Lan Gong (Reviewer #2)

Transaction Report:

DOI: <https://doi.org/10.1128/spectrum.05066-22>

January 3, 2023

Dr. Justine W Debelius
Johns Hopkins University Bloomberg School of Public Health
Department of Epidemiology
Baltimore, MD

Re: Spectrum05066-22 (The local tumor microbiome is associated with survival in late-stage colorectal cancer patients)

Dear Dr. Justine W Debelius:

Link Not Available

Sincerely,

Zhenjiang Xu

Journals Department
Reviewer comments:

Reviewer #1 (Comments for the Author):

General comments:

- Please consider having the ms (manuscript) reviewed by an English editing service or several English speakers.
- Check the overall format, including spacing, indentation and etc
- Good experimental plan and analyses.
- Improve the delivery of the ms, eg the choice of terms and by providing more details, will significantly improve the quality of the whole ms.
- Use a formal tone in your ms.
- Rarefaction curve and Alpha diversity are not discussed in this ms. Please justify.

Specific comments:

Line Comment

4 Potential area of research

51-52 Please explain in a greater detail regarding broad pattern in the observed microbiome. What was consistent between your results and the previous studies?

52 Observed

55 Similar characteristics - Do you mean "same cancer stage or disease status"? more details are needed.

63 Sample's position? Please clarify

62-64 Unclear, please improve the sentence - "We did not find a statistically significant association between a sample's position in CTF space and survival (unadjusted permanova $R^2=0.012$; $p=0.296$, 999 permutation, Figure S3, Table S2)."

The use of "space" is informal and confusing. Apply to all.

64 Are the differences significant? Please provide the significance details.

82-83 Suggest using a more formal tone in your ms.

86 Abundance of what?

86 Greater change

88 Improve the sentence

96, 133 Why the use of "/" between Escherichia/Shigella? They are different genus and are not interchangeable.

98-99 Unclear interpretation. Please consider explain these in separate sentences and provide more details to each observation.

100 Which 2 groups? Between survival outcomes or between tissue types? Please do not leave out these important details even though it can be repetitive.

108 Use a more accurate description than "larger"

112 Do you mean profound?

140 Feature is not the right use of word here, please be more specific.

144 Important statement. Please improve the delivery of this sentence. What are the tissue types? Where were those tissue isolated from?

151, 156 "let alone" is a wrong choice of term in this professional paper. It does not directly reflect

156-157 Like the past studies, we observed and described a strong intraindividual similarity - this sentence needs citation.

160-161 It sounds like the strong intra-individual similarity you showed is because of the statistical model. Is this what you want to convey to the audience?

161-163 Need citation on how you assume this possibility?

182-183 Please explain more in detailed how the different sequencing methods may contribute the different observation of the relative abundances.

183-192 The author explained their results were similar to the results from a Chinese cohort. Please discuss how the increased or decreased abundances of each genus or collectively mentioned is related to the survival outcomes with literature supports.

195 "at the time of surgery" - is this a consistent information for all the patients recruited in this study? did they all have the surgery at the time frame, eg at which stage of the cancer? Please clarify. For future reference, when (eg at which stage of cancer) would you suggest for the tumour tissue samples to be isolated?

235 Patients' details or information? Apply to all

236 Survival outcomes?

241-242 Delete or improve the delivery of this sentence. "There was no difference in the survival status in the samples with insufficient sequencing depth.

262 Paired tumor and normal tissue samples - please mention the type of tissue

272 Do you mean an average length of the trimmed sequences? Or do you mean all the reads were trimmed to the exact same size? Please clarify.

273 What tool did you use to join the reads? DADA2?

307 Which

308 Survival outcomes (apply to all)

354 Hanging sentence?

Figures - Provide a figure label to every sub figure, to clearly indicate the variables you are presenting in each one of them.

Reviewer #2 (Comments for the Author):

In this study, Debelius et al. investigated the microbiome of paired tumor and adjacent tissue samples from late-stage colorectal cancer patients, and identified clear differences in the microbiome between normal and tumor tissue. The authors also found that a larger difference between tissue types was associated with poor prognosis and that the local tumor microbiome linked to

survival. This is a carefully conducted study with fine quality in manuscript-preparing and logic. The experiments are straightforward, and the data are generally well presented in a clear manner. I have a few comments and suggestions as follow.

1) Since gut/fecal microbiome is associated with survival in colorectal cancer. It will be great if the authors can include fecal microbiome data and analysis its correlation with tissue microbiome in terms of prognosis.

2) Although the authors already listed as one of their study limitations (line 212), lower sequencing depth with 16S rRNA amplicon sequencing limited microbiome data (genus level) especially for the survival-associated tumor microbiome profile. Shot-gun metagenome sequencing of 10-20 tumor samples will provide species/strain level data with possible real functional analysis.

3) Also, the authors realized lack of validation cohort as a study limitation (line 209). To solve this problem without adding more subjects, the authors can re-analysis the data from other published studies to validate their model. They actually already suggested this in the manuscript (line 161-162): "Re-analysis of prior publications using subject aware methods may identify the same patterns we found: strong individual microbial signatures with a difference between the tissue types". Could the authors perform the reanalysis and prove their hypothesis?

4) Same as the above point, re-analysis can be performed to validate the survival-associated tumor microbiome with the data from other studies of Irish patients (line 172), since "It is likely this disagreement is due to differences in methods used for differential abundance" (line 182-183).

Staff Comments:

Preparing Revision Guidelines

Please return the manuscript within 60 days; if you cannot complete the modification within this time period, please contact me. If you do not wish to modify the manuscript and prefer to submit it to another journal, please notify me of your decision immediately so that the manuscript may be formally withdrawn from consideration by Microbiology Spectrum.

General comments:

- Please consider having the ms (manuscript) reviewed by an English editing service or several English speakers.
- Check the overall format, including spacing, indentation and etc
- Good experimental plan and analyses.
- Improve the delivery of the ms, eg the choice of terms and by providing more details, will significantly improve the quality of the whole ms.
- Use a formal tone in your ms.
- Rarefaction curve and Alpha diversity are not discussed in this ms. Please justify.
-

Specific comments:

Line	Comment
4	Potential area of research
51-52	Please explain in a greater detail regarding broad pattern in the observed microbiome. What was consistent between your results and the previous studies?
52	Observed
55	Similar characteristics - Do you mean “same cancer stage or disease status”? more details are needed.
63	Sample’s position? Please clarify
62-64	Unclear, please improve the sentence – “We did not find a statistically significant association between a sample’s position in CTF space and survival (unadjusted permanova $R^2=0.012$; $p=0.296$, 999 permutation, Figure S3, Table S2).” The use of “space” is informal and confusing. Apply to all.
64	Are the differences significant? Please provide the significance details.
82-83	Suggest using a more formal tone in your ms.
86	Abundance of what?
86	Greater change
88	Improve the sentence
96, 133	Why the use of “/” between Escherichia/Shigella? They are different genus and are not interchangeable.
98-99	Unclear interpretation. Please consider explain these in separate sentences and provide more details to each observation.
100	Which 2 groups? Between survival outcomes or between tissue types? Please do not leave out these important details even though it can be repetitive.
108	Use a more accurate description than “larger”
112	Do you mean profound?
140	Feature is not the right use of word here, please be more specific.
144	Important statement. Please improve the delivery of this sentence. What are the tissue types? Where were those tissue isolated from?
151, 156	“let alone” is a wrong choice of term in this professional paper. It does not directly reflect
156-157	Like the past studies, we observed and described a strong intraindividual similarity – this sentence needs citation.
160-161	It sounds like the strong intra-individual similarity you showed is because of the statistical model. Is this what you want to convey to the audience?

161-163	Need citation on how you assume this possibility?
182-183	Please explain more in detailed how the different sequencing methods may contribute the different observation of the relative abundances.
183-192	The author explained their results were similar to the results from a Chinese cohort. Please discuss how the increased or decreased abundances of each genus or collectively mentioned is related to the survival outcomes with literature supports.
195	“at the time of surgery” – is this a consistent information for all the patients recruited in this study? did they all have the surgery at the time frame, eg at which stage of the cancer? Please clarify. For future reference, when (eg at which stage of cancer) would you suggest for the tumour tissue samples to be isolated?
235	Patients’ details or information? Apply to all
236	Survival outcomes?
241-242	Delete or improve the delivery of this sentence. “There was no difference in the survival status in the samples with insufficient sequencing depth.
262	Paired tumor and normal tissue samples – please mention the type of tissue
272	Do you mean an average length of the trimmed sequences? Or do you mean all the reads were trimmed to the exact same size? Please clarify.
273	What tool did you use to join the reads? DADA2?
307	Which
308	Survival outcomes (apply to all)
354	Hanging sentence?
Figures	Provide a figure label to every sub figure, to clearly indicate the variables you are presenting in each one of them.

Dear Professor Xu,

We wish to thank the reviewers, especially reviewer 2, for their constructive comments. We were concerned about reviewer 1's focus on style over substance. We believe peer review should be about scientific content, rather than questions of tone or formality. We have responded to reviewer 1's critiques we determined to be scientific and added the note "this is stylistic" to points we believe to reflect tone, rather than content.

Sincerely,

Renate Slind Olsen and Justine Debelius

Reviewer comments:

Reviewer #1 (Comments for the Author):

General comments:

- Please consider having the ms (manuscript) reviewed by an English editing service or several English speakers.

This is stylistic.

- Check the overall format, including spacing, indentation and etc

Microbiology Spectrum has a format free initial submission.

(see <https://journals.asm.org/format-neutral-submissions?journal=spectrum>).

- Good experimental plan and analyses.

Thank you.

- Improve the delivery of the ms, eg the choice of terms and by providing more details, will significantly improve the quality of the whole ms.

It is unclear what is meant by this comment; it appears to be stylistic.

- Use a formal tone in your ms.

This is stylistic.

- Rarefaction curve and Alpha diversity are not discussed in this ms. Please justify.

Our primary findings are based on rarefaction-less metrics. Compositional tensor factorization (CTF), robust principal components analysis (rPCA), and differential ranking (DR) are all compositional techniques which are performed on unrarefied data (1–4). We describe the filtering approach we applied before each of these techniques in lines 311-312, 319-321, and 336-338, respectively. Two supplemental analyses were performed on rarefied data. However, given the work by Weiss et al (5) demonstrating that the relationship between samples remains consistent for a common rarefaction depth, a rarefaction curve is unnecessary.

We have included alpha diversity in the supplemental analysis, presented on GitHub (6).

Specific comments:

Line Comment

4 Potential area of research

Changed

51-52 Please explain in a greater detail regarding broad pattern in the observed microbiome. What was consistent between your results and the previous studies?

It is not clear how a very high level text description of what can be seen in the figure would improve the manuscript. The figure is provided primarily to demonstrate that our samples look like an adult gut. We recommend Huttenhower et al (6) as a useful high level overview of western adult body sites, as well as the referenced article on the Swedish microbiome (7).

52 Observed

Changed, although the overserved microbiome leads to lots of interesting questions.

55 Similar characteristics - Do you mean "same cancer stage or disease status"? more details are needed.

These are provided in the figures and methods (See the for figure S2; lines 426-429):

Paired distances were extracted as the distance between an individual's tumor and adjacent normal tissue. Interindividual distance was compared to the interindividual distance to samples of the same tissue type, anatomical location, and survival group

63 Sample's position? Please clarify

62-64 Unclear, please improve the sentence - "We did not find a statistically significant association between a sample's position in CTF space and survival (unadjusted permanova $R^2=0.012$; $p=0.296$, 999 permutation, Figure S3, Table S2)."

The use of "space" is informal and confusing. Apply to all.

"Ordination space" is a widely used description for the relationship between samples projected in constrained dimensions (8, 9). Online training resources that are used to teach the concept also use the same language: the popular GUSTAME website (10) and QIIME 2 documentation (11), describe the dimensionally reduced relationship between samples as "space". We have discussed this language with colleagues at our institutions and others, none had better suggestions that efficiently captured the relationship between samples in a clear, concise way. With this in mind, we have re-written lines 60-62 to clarify that CTF is an ordination technique, and added an additional sentence:

This analysis projects high dimensional microbiome data into a three-dimensional ordination space, relating samples and their component features.

We have also corrected the sentence on lines 62-64, replacing "sample" with "subject" since this more accurately reflects the observed data. Beyond this correction, we do not know how to make the statement more clear. We have presented a relationship between exposure (subject position in CTF space) and the outcome (survival), along with the appropriate effect size measurement, p-value description, and a description of where further details can be found.

64 Are the differences significant? Please provide the significance details

Please see figure 1A-D. However, we have added a note about significance on line 67:

Paired samples from the same individual showed consistent, directional differences, primarily along principal component (PC) 2 and PC 3 (Figure 1A-D; permutative paired sample t-test $p=0.001$, 999 permutations for all PCs).

82-83 Suggest using a more formal tone in your ms.

This is stylistic.

86 Abundance of what?

We have re-written this as "abundant features" (now lines 105-106)

86 Greater change

Now line 107

88 Improve the sentence

This is stylistic and ambiguous.

96, 133 Why the use of "/" between *Escherichia/Shigella*? They are different genus and are not interchangeable.

Members of genus *Shigella* are, in fact, *Escherichia coli*, which recognized for more than 20 years (12, 13). They cannot be differentiated by 16S rRNA sequencing, and databases based on larger panels of marker genes suggest this is a broader challenge (14). Databases have selected a variety schema to represent this taxonomic ambiguity; the Silva database we used has selected the genus level assignment of "*Escherichia/Shigella*". It would be disingenuous to change this representation. Re-naming features mapped to this genus to the family level would imply our taxonomic classification was less precise than it was. It might also suggest that we identified other members of family *Enterobacteraceae*, which would be both imprecise and inaccurate. Substituting the slash symbol for a written conjunction would suggest the identification of more features than are represented in our dataset. Altering the annotation in some other way could suggest a broader, manual curation of the database, which has consequences for external validity. In short, from a 16S rRNA sequencing perspective, *Escherichia/Shigella* are a single, interchangeable genus, and alternative annotation would be imprecise, inaccurate, and disingenuous.

98-99 Unclear interpretation. Please consider explain these in separate sentences and provide more details to each observation.

We are unclear how to interpret this comment. The results are as described: the ratio of taxa is higher in tumor samples for both survival groups, and the overall ratio is higher in short term survivors. This is also demonstrated in figure 2F.

100 Which 2 groups? Between survival outcomes or between tissue types? Please do not leave out these important details even though it can be repetitive.

Survival types. (see Line 122-123). Also, very much stylistic.

108 Use a more accurate description than "larger"

This is stylistic

112 Do you mean profound?

We have used the word pronounced

140 Feature is not the right use of word here, please be more specific.

This is stylistic; "feature" has been used throughout the manuscript.

144 Important statement. Please improve the delivery of this sentence. What are the tissue types? Where were those tissue isolated from?

This comment is confusing. The tissue types have been consistently described through out the manuscript: tumor and adjacent normal tissue. The specific biopsy locations (left colon, right colon, and rectum) are provided in table S1.

151, 156 "let alone" is a wrong choice of term in this professional paper. It does not directly reflect

This is stylistic

156-157 Like the past studies, we observed and described a strong intraindividual similarity - this sentence needs citation.

We have joined this and the subsequent sentence citing works with a colon to clarify the two statements belong together. (Lines 180-183)

160-161 It sounds like the strong intra-individual similarity you showed is because of the statistical model. Is this what you want to convey to the audience?

The strong intraindividual similarity is a fundamental property of the microbiome, as demonstrated throughout the paper. However, we reversed the order of the sentence. (Lines 188-189)

161-163 Need citation on how you assume this possibility?

We included a replication cohort at the request of Reviewer 2 (see lines 80-97, 185-189, 367-390, and supplemental Figure 4).

182-183 Please explain more in detailed how the different sequencing methods may contribute the different observation of the relative abundances.

Presumably this refers to the question of differential abundance producing different results, rather than asking for a full description of how methodological approaches result in differences in 16S rRNA studies. Nearing et al (15) recently demonstrated that differential abundance do not identify consistent sets of features on the same dataset. However, this comparison is further complicated because neither differential abundance approaches discussed here were included in the Nearing comparison. Our approach (described in lines 323-338) used a compositionally aware ordination to identify features responsible for the greatest separation between samples. In contrast, Flemer et al constructed groups through cross correlation followed by clustering to construct their clade groups which they named based on representative features (16). This approach was not compositionally aware, increasing the probability of spurious correlations (17). Our manuscript has already demonstrated that differences in analysis can directly change inference; building on this and the existing literature around differential abundance it is highly likely at least a partial explanation.

183-192 The author explained their results were similar to the results from a Chinese cohort. Please discuss how the increased or decreased abundances of each genus or collectively mentioned is related to the survival outcomes with literature supports.

It is unclear how this would improve the quality of the paper or scientific rigor of the field. We introduced qPCR-based markers in our introduction (lines 9-11); then we discussed how organisms in isolation fail to reflect the biological reality of the system (lines 13-18). One of the motivations for our work was the lack of community-centric approaches to microbiome survival. We made a conscientious choice not to analyze organisms in isolation and instead used data driven, polymicrobial approaches to look for features contributing to survival. It would be antithetical to the choices we have made analytically and narratively to then discuss organisms in isolation. There are not good complex culture-based studies to understand how these organisms interact, and anything we present would be hand-waving speculation at best and irresponsible overselling of our results at worst.

195 "at the time of surgery" - is this a consistent information for all the patients recruited in this study? did they all have the surgery at the time frame, eg at which stage of the cancer? Please clarify. For future reference, when (eg at which stage of cancer) would you suggest for the tumour tissue samples to be isolated?

See lines 256-626 which describe full recruitment procedures. Cancer stage was determined at the time of surgery. As all our samples are late stage CRC, it is not possible to determine which cancer stage would be appropriate to determine mortality.

235 Patients' details or information? Apply to all

This is stylistic.

236 Survival outcomes?

This is stylistic, and our model has only one outcome.

241-242 Delete or improve the delivery of this sentence. "There was no difference in the survival status in the samples with insufficient sequencing depth.

No, thank you. It says exactly what it is intended to say.

262 Paired tumor and normal tissue samples - please mention the type of tissue

We introduce the fact that we are studying the intestinal microbiome in lines 30-35 and then go on to demonstrate we have a gastrointestinal microbiome in Figure S1. We specify that samples were taken from multiple locations along the GI tract (Table S1).

272 Do you mean an average length of the trimmed sequences? Or do you mean all the reads were trimmed to the exact same size? Please clarify.

273 What tool did you use to join the reads? DADA2?

The forward and reverse trim length for dada2 are standard processing parameters. The algorithm first trims the forward and reverse reads (separately) to the same length. A quality filter is applied. The run specific sequencing error rate is determined using a machine learning model, and this error model is used to correct (denoise) the remaining reads. Dada2 then joins the denoised reads and performs chimera removal. All of these are steps within the DADA2 pipeline, as described in the DADA2 paper (18), DADA2 tutorials (<https://benjjneb.github.io/dada2/tutorial.html>), and clearly indicated in our code (https://github.com/ctmrbio/Amplicon_workflows).

307 Which

Corrected

308 Survival outcomes (apply to all)

This is stylistic, and perhaps more importantly, there is only one outcome.

354 Hanging sentence?

Corrected

Figures - Provide a figure label to every sub figure, to clearly indicate the variables you are presenting in each one of them.

It is unclear what this means. We have clearly added a label for each panel as appropriate (in some cases, a panel may include multiple axes). Axes are carefully and clearly cross labeled, and captions specify adjustments and model types.

Reviewer #2 (Comments for the Author):

In this study, Debelius et al. investigated the microbiome of paired tumor and adjacent tissue samples from late-stage colorectal cancer patients, and identified clear differences in the microbiome between normal and tumor tissue. The authors also found that a larger difference between tissue types was associated with poor prognosis and that the local tumor microbiome linked to survival. This is a carefully conducted study with fine quality in manuscript-preparing and logic. The experiments are straightforward, and the data are generally well presented in a clear manner. I have a few comments and suggestions as follow.

Thank you for your kind words.

1) Since gut/fecal microbiome is associated with survival in colorectal cancer. It will be great if the authors can include fecal microbiome data and analysis its correlation with tissue microbiome in terms of prognosis.

We agree this would be a fascinating study, and hope others designing future experiments are able to collect fecal samples. However, we are working within the limits of a retrospective study initiated in 1996. No fecal samples were collected.

2) Although the authors already listed as one of their study limitations (line 212), lower sequencing depth with 16S rRNA amplicon sequencing limited microbiome data (genus level) especially for the survival-associated tumor microbiome profile. Shotgun metagenome sequencing of 10-20 tumor samples will provide species/strain level data with possible real functional analysis.

Again, while an interesting suggestion, it is unclear how shotgun sequencing of a subset of samples will substantially improve our inference. There are three issues at play here.

First, it is unclear how to select this magical subset of 10-20 samples for improved inference. Our results clearly show strong individual microbiome profiles (lines 53-55; Figure S2), how would the reviewer advise selecting the samples? Should we pick the most extreme based on the 16S rPCA? Some other criterion? How do we know we will have sufficient statistical power for this comparison? Next, we present data at the ASV level – a nucleotide accurate barcode for each organism, although we do use genus level annotation to facilitate biological inference and to make the information more useful to a wide audience. Work by Armour et al (19) suggests that taxonomic inference provided at the family level is sufficient to distinguish colorectal cases from controls; our ASVs provide significantly more specificity. In our limitations, we specify, “Without robust functional prediction and the ability to assemble genome units, we are limited in our mechanistic insight.” This is not simply a critique of 16S, but a problem with functional annotation in contemporary microbiome research. Short read functional annotation pipelines like HUMANN and Superfocus offer an overall catalogue of genes, but do not offer discrete functional units (20, 21). Our best option for useful genome assembly would be a cross linking technology like Hi-C, however, this must be done before sample extraction and we no longer have primary sample left. Finally, our samples come from biopsies. The high degree of human contamination in these samples makes it difficult to obtain high quality metagenomic results at an affordable price. A published attempt at metagenomics in similar samples worked with a rarefaction depth of 200 reads due to their inability to retain reads from microbial DNA (22). Substantial work – likely sufficient for a stand-alone publication – would be required to validate the methods for this extraction. Due to the high heterogeneity, minimal improvement in taxonomic inference, poor functional annotation, and high cost, it is our evaluation that the cost-to-benefit ratio of metagenomic sequencing on even a few samples is too high.

3) Also, the authors realized lack of validation cohort as a study limitation (line 209). To solve this problem without adding more subjects, the authors can re-analysis the data from other published studies to validate their model. They actually already suggested this in the manuscript (line 161-162): "Re-analysis of prior publications using subject aware methods may identify the same patterns we found: strong individual microbial signatures with a difference between the tissue types". Could the authors perform the reanalysis and prove their hypothesis?

It would not be feasible to add more subjects to this study from our existing population. A new cohort would have to be an independent study. However, we tried to use the Flemer cohort as a validation set (16, 23). We selected this cohort since the data is publicly available and they used the same 314F starting position that we used. We were able to link the metadata for a subset of samples, which we re-analyzed to address questions of the difference between sample types (point 3). We present these results in lines 80-93 and Figure S4. Briefly, a permanova test for global differences does not find a difference between tumor tissue in either our data or the replication cohort, however, the subject aware CTF ordination identified a consistent directional difference in the validation set. We were also able to show a statistically significant difference associated with our tumor/normal tissue ALR.

4) Same as the above point, re-analysis can be performed to validate the survival-associated tumor microbiome with the data from other studies of Irish patients (line 172), since "It is likely this disagreement is due to differences in methods used for differential abundance" (line 182-183).

However, it was not possible to re-analyze the data for survival, as this information was not provided for the cohort and we could not find another cohort with the correct forward primers for replication. While this may decrease confidence in our specific findings, we believe it highlights the importance of this study and it's corresponding publicly available data since there are not other accessible comparable data sets.

References

1. Martino C, Morton JT, Marotz CA, Thompson LR, Tripathi A, Knight R, Zengler K. 2019. A Novel Sparse Compositional Technique Reveals Microbial Perturbations. *mSystems* 4.
2. Martino C, Shenhav L, Marotz CA, Armstrong G, McDonald D, Vázquez-Baeza Y, Morton JT, Jiang L, Dominguez-Bello MG, Swafford AD, Halperin E, Knight R. 2021. Context-aware dimensionality reduction deconvolutes gut microbial community dynamics. *Nature Biotechnology* 39:165–168.
3. Morton JT, Marotz C, Washburne A, Silverman J, Zaramela LS, Edlund A, Zengler K, Knight R. 2019. Establishing microbial composition measurement standards with reference frames. *Nat Commun* 10:2719.
4. Morton JT, Jin D-M, Mills RH, Shao Y, Rahman G, Berding K, Needham BD, Zurita MF, David M, Averina OV, Kovtun AS, Noto A, Mussap M, Wang M, Frank DN, Li E, Zhou W, Fanos V, Danilenko VN, Wall DP, Cárdenas P, Baldeón ME, Xavier RJ, Mazmanian SK, Knight R, Gilbert JA, Donovan SM, Lawley TD, Carpenter B, Bonneau R, Taroncher-Oldenburg G. 2022. Multi-omic analysis along the gut-brain axis points to a functional architecture of autism. *bioRxiv* <https://doi.org/10.1101/2022.02.25.482050>.

5. Weiss S, Xu ZZ, Peddada S, Amir A, Bittinger K, Gonzalez A, Lozupone C, Zaneveld JR, Vázquez-Baeza Y, Birmingham A, Hyde ER, Knight R. 2017. Normalization and microbial differential abundance strategies depend upon data characteristics. *Microbiome* 5:27.
6. Debelius J. 2023. ctmrbio/crc-survival: Resubmission release. Zenodo.
7. Human Microbiome Project Consortium. 2012. Structure, function and diversity of the healthy human microbiome. *Nature* 486:207–214.
8. Hugerth LW, Andreasson A, Talley NJ, Forsberg AM, Kjellström L, Schmidt PT, Agreus L, Engstrand L. 2020. No distinct microbiome signature of irritable bowel syndrome found in a Swedish random population. *Gut* 69:1076–1084.
9. Legendre P, Legendre, Louis. 2012. Chapter 9. Ordinations in Reduced Space, p. 425–520. *In* Numerical Ecology, 3rd ed. Elsevier, Amsterdam.
10. Ramette A. 2007. Multivariate analyses in microbial ecology. *FEMS Microbiology Ecology* 62:142–160.
11. Principal coordinates analysis - GUSTA ME.
<https://sites.google.com/site/mb3gustame/dissimilarity-based-methods/principal-coordinates-analysis>. Retrieved 28 February 2023.
12. QIIME 2 Development team. 2023. PD Mouse Tutorial. QIIME 2 docs.
<https://docs.qiime2.org/2022.11/tutorials/pd-mice/>. Retrieved 28 February 2023.
13. Lan R, Reeves PR. 2002. Escherichia coli in disguise: molecular origins of Shigella. *Microbes Infect* 4:1125–1132.
14. Abram K, Udaondo Z, Bleker C, Wanchai V, Wassenaar TM, Robeson MS, Ussery DW. 2021. Mash-based analyses of Escherichia coli genomes reveal 14 distinct phylogroups. 1. *Commun Biol* 4:1–12.
15. Parks DH, Chuvochina M, Reeves PR, Beatson SA, Hugenholtz P. 2021. Reclassification of Shigella species as later heterotypic synonyms of Escherichia coli in the Genome Taxonomy Database. bioRxiv <https://doi.org/10.1101/2021.09.22.461432>.
16. Nearing JT, Douglas GM, Hayes MG, MacDonald J, Desai DK, Allward N, Jones CMA, Wright RJ, Dhanani AS, Comeau AM, Langille MGI. 2022. Microbiome differential abundance methods produce different results across 38 datasets. 1. *Nat Commun* 13:342.
17. Flemer B, Lynch DB, Brown JMR, Jeffery IB, Ryan FJ, Claesson MJ, O'Riordain M, Shanahan F, O'Toole PW. 2017. Tumour-associated and non-tumour-associated microbiota in colorectal cancer. *Gut* 66:633–643.
18. Gloor GB, Macklaim JM, Pawlowsky-Glahn V, Egozcue JJ. 2017. Microbiome Datasets Are Compositional: And This Is Not Optional. *Front Microbiol* 8:2224.
19. Callahan BJ, McMurdie PJ, Rosen MJ, Han AW, Johnson AJA, Holmes SP. 2016. DADA2: High-resolution sample inference from Illumina amplicon data. 7. *Nature Methods* 13:581–583.

20. Armour CR, Topçuoğlu BD, Garretto A, Schloss PD. 2022. A Goldilocks Principle for the Gut Microbiome: Taxonomic Resolution Matters for Microbiome-Based Classification of Colorectal Cancer. *mBio* e0316121.
21. Beghini F, McIver LJ, Blanco-Míguez A, Dubois L, Asnicar F, Maharjan S, Mailyan A, Manghi P, Scholz M, Thomas AM, Valles-Colomer M, Weingart G, Zhang Y, Zolfo M, Huttenhower C, Franzosa EA, Segata N. 2021. Integrating taxonomic, functional, and strain-level profiling of diverse microbial communities with bioBakery 3. *eLife* 10:e65088.
22. Silva GGZ, Green KT, Dutilh BE, Edwards RA. 2016. SUPER-FOCUS: a tool for agile functional analysis of shotgun metagenomic data. *Bioinformatics* 32:354–361.
23. Debesa-Tur G, Pérez-Brocal V, Ruiz-Ruiz S, Castillejo A, Latorre A, Soto JL, Moya A. 2021. Metagenomic analysis of formalin-fixed paraffin-embedded tumor and normal mucosa reveals differences in the microbiome of colorectal cancer patients. 1. *Scientific Reports* 11:391.
24. Cronin P, Murphy CL, Barrett M, Ghosh TS, Pellanda P, O'Connor EM, Zulquernain SA, Kileen S, McCourt M, Andrews E, O'Riordain MG, Shanahan F, O'Toole PW. 2022. Colorectal microbiota after removal of colorectal cancer. *NAR Cancer* 4:zcac011.

March 27, 2023

Dr. Justine W Debelius
Johns Hopkins University Bloomberg School of Public Health
Department of Epidemiology
Baltimore, MD

Re: Spectrum05066-22R1 (The local tumor microbiome is associated with survival in late-stage colorectal cancer patients)

Dear Dr. Justine W Debelius:

Your manuscript has been accepted, and I am forwarding it to the ASM Journals Department for publication. You will be notified when your proofs are ready to be viewed.

Sincerely,

Zhenjiang Xu
Editor, Microbiology Spectrum

Journals Department
General comment:

As I mentioned in my first review, this is an excellent scientific paper with a great experimental plan and fantastic findings. I am pleased to have read the authors' passionate work on science. Nonetheless, I believe that science communication also holds a crucial role in sharing knowledge effectively. As journal articles serve as platforms for science communication and knowledge sharing, it is crucial to work on communication so that the knowledge is accessible to a broader audience.

I understand that some of my suggestions may not have been well-received by the authors, but I am happy to see some changes in the words made by the authors which have improved the clarity and quality of the ms. And I hope to emphasize that my intention was not to offend or criticize their work. Rather, my aim was to offer constructive feedback to enhance the clarity and accessibility of their findings. And for the authors' information, it is the reviewers' job to ensure the ms is professionally written and comprehensive.

Ultimately, I respect the authors' standards in the delivery of their great work, and I appreciate their commitment to scientific excellence. I believe that open and respectful communication is fundamental in advancing scientific research and knowledge sharing, and I look forward to more great works from the authors.

Well done.